# Extensible LED-Induced Integrated Fluorescence Detection Module for Quantitative Analysis of Lucigenin Concentration

Xiaoguang Qi [1,*], Xianglong Hao [1], Muzi Zhang [1], Lili Jiang [1], Wenyue Gao [1] and Chi Wu [1,2,3,*]

1  Shandong Provincial Center for In-Situ Marine Sensors, Institute of Marine Science and Technology, Shandong University, Qingdao 266237, China
2  Southern Marine Science and Engineering Guangdong Laboratory (Guangzhou), Guangzhou 511458, China
3  Shandong Provincial Center for In-Situ Marine Sensors, Aixsensor Co., Dezhou 253000, China
*  Correspondence: qi2020@sdu.edu.cn (X.Q.); qi.wu@sdu.edu.cn (C.W.)

**Abstract:** We developed an extensible LED-induced fluorescence detection module with a highly integrated and ultra-compact structure. A target-oriented design methodology was used to demonstrate the module's optimal design. Lucigenin solution was used as a test sample in evaluation trials to demonstrate the module's quantitative fluorescence detection capability. Results showed that the integrated module has an outstanding linear response in the range of 0–1 $\mu mol \cdot L^{-1}$, with sensitivity and limit of detection (LOD) of 0.1692 $V/\mu mol \cdot L^{-1}$ and 0.03 $\mu mol \cdot L^{-1}$, respectively. Statistical analyses showed that our integrated module has extremely high repeatability and accuracy, i.e., the values of Pearson's correlation coefficient and root-mean-square error exceeded 0.9995 and 1.8‰, respectively. More importantly, the integrated module possesses favorable extensibility and can realize on-demand rapid fluorescence-signal detection of other targets using appropriate parameter combinations. This module offers new opportunities for reliable, cost-effective and easy-to-use fluorescence-signal detection, especially in resource-constrained fluorescence detection applications.

**Keywords:** extensible; LED-induced; ultra-compact; fluorescence detection; lucigenin concentration; quantitative analysis

## 1. Introduction

The fluorescence quantitative analysis method (FQAM) is a versatile technological means of analyzing substances [1]. As is well known, many substances have endogenous fluorophores—or can be conjugated to a fluorescent reagent—and when activated by excitation light at a specific wavelength, emit fluorescence [2]. Thus, FQAM has been applied in various fields, including pharmacology [3], biology [4], physiology [5,6], and environmental sciences [7–9]. The measuring devices used for this application include fluorescent illuminometer (FI) [4], fluorescence spectrometer (FS) [1,7] and other special instruments [3,8,9]. Although these devices have shown remarkable capabilities in analyzing small sample volumes with high sensitivity and low detection limits [10,11], their adoption has been limited outside of research laboratories, especially in in situ detection. Potential barriers to in situ detection include the cost and complexity of the required optical system, the bigger size, the short lifespan (200–300 h), and the requirement for professional personnel to conduct the analyses to obtain reliable results [12].

Researchers have proposed various fluorescence detection schemes in recent years to enable the application of FQAM in more situations. For example, Yang Jinlan et al. developed a fluorescent polyethyleneimine-protected copper nanoclusters (PEI-CuNCs) probe for quantitative analysis of biothiols and acetylcholinesterase (AChE) [4]. They found that two biothiols, glutathione and cysteine, were detected with LODs of 0.26 M and 0.34 M within linear ranges of 0.5–25 M and 1–25 M, respectively. Cui Yaoyao et al. built an excitation emission fluorescence matrix for four categories of polycyclic aromatic hydrocarbons and parallel factor framework-clustering analysis (PFFCA) using FS [7]. They

found that the specificity of this method is as high as 96.7% for six types of oil samples. However, the size and cost of FI and FS, the complex preparation process and the frequent bulb replacement required still restrict the application of FQAM outside the laboratory. Thus, several self-developed devices have been proposed for online or in vivo application. For instance, Martin Brandl et al. developed a portable fluorescence sensor system for the analysis of dissolved organic matter (DOM) [8]. Li Wanxiang et al. built a detection scheme using a linear CCD for simultaneous measurement of absorbance and fluorescence of chlorophyll-a [9]. They used the CCD integration time transformation method to quantitatively analyze chlorophyll-a concentration in natural water and reduced the minimum detection limit from 0.01 µg/L to 0.0025 µg/L in the detection range of 0.0025–130 µg/L. Heykel et al. developed a compact fiber fluorescence detection device [13]. They mounted a polystyrene microsphere at the end of the fiber and achieved single molecule sensitivity and remote detection capabilities. Yu-Chung et al. developed a fiber fluorescence detection system [14]. They used a dual-clad optical fiber to excite fluorescent nanosphere samples and achieved ultrasensitive two-photon fluorescence detection in neoplastic cells for early screening and treatment in vivo. These self-developed devices have shown powerful practicability in online or in vivo quantitative fluorescence detection. However, these devices are complicated, time consuming, and require sampling for laboratory measurement and analysis, which are not suitable for in situ fluorescence detection. In addition, the low coupling efficiency and higher alignment requirement between the fiber core and spatial light beam has been a significant barrier to further lowering prices and compressing size.

Light-emitting diodes (LEDs), which have various advantages including being extremely compact, cost-effective, energy-efficient and power-stable, have the potential to facilitate the multi-function, integration, and miniaturization of the fluorescence quantitative analysis system. LEDs are semiconductor-based light sources in which the bandgap (eV) affects the energy and wavelength of the light emitted, and which may suit the excitation criteria of most fluorescent molecules [15]. For example, LEDs have been used as excitation light sources when developing absorption- [16,17], fluorescence- [18] and reflection- [19] spectroscopy-based optic-analytical instruments for various analytical application. These LED-induce fluorescence detection schemes have promoted the development of a comprehensive environmental surveillance system that can more strictly control water pollution and track changes in marine ecology. Therefore, we have reason to believe that the LED-induced scheme will promote the miniaturization and integration of FQAM techniques and solve the cost and power consumption problems.

In this study, we developed an extensible LED-induced fluorescence detection module. A target-oriented design methodology based on the 3D fluorescence spectrum was built by adjusting the LED's central wavelength and narrowband filter parameters to determine the concentrations of fluorescence molecules. Full performance evaluation trials using lucigenin as a fluorescent indicator demonstrated that the integrated module has an outstanding linear response in the range of 0–1 $\mu mol \cdot L^{-1}$, with sensitivity and limit of detection (LOD) of, respectively, 0.1692 $V/\mu mol \cdot L^{-1}$ and 0.03 $\mu mol \cdot L^{-1}$. Pearson's correlation coefficient (R), root-mean-square error (RMSE), and mean absolute percentage error (MAPE) of the multiple-sample measurements exceeded 0.9995, 1.8‰, and 5.079%, respectively, based on statistical analysis of the data. In conclusion, automatic inspection can benefit from the LED-induced integrated module's higher linearity, repeatability and stability, as well as its modular and expandable architecture.

## 2. Theoretical Method and Device

### 2.1. Measurement Analytic Theory

In our module setup, the excitation light ($\lambda_{EX}$) from the LED is projected into a sample solution, as shown in Figure 1a. The sample region irradiated by the excitation light is defined as the detecting volume $V$. Part of the excitation light is absorbed by the sample solution and the remainder is scattered. The fluorophore absorbs the photon energy and jumps to an excited state when the sample region is irradiated by the excitation light. The

fluorophore in the sample solution subsequently emits fluorescence when it returns to the ground state.

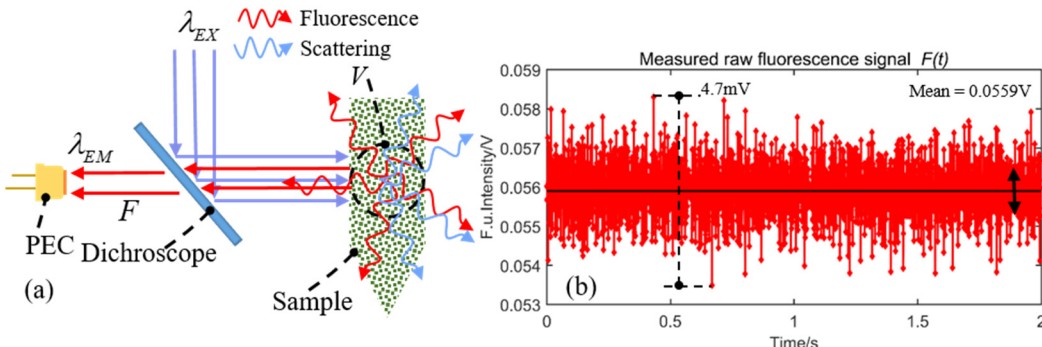

**Figure 1.** (**a**) Schematic view of fluorescence excitation. PEC: Photo-Electric Converter. (**b**) Raw fluorescence signal $F(t)$ received by the PEC at 0.4 µmol/L.

The fluorescence intensity of the solution $F$ is related to the solution's absorption intensity $I_a$ and the fluorophore's fluorescence quantum yield $\beta$.

$$F = \beta I_a = \beta(I_0 - I_t) \tag{1}$$

where $I_0$ is the intensity of excitation light and $I_t$ is the intensity of transmitted light. Based on the Lambert-Beer law [20],

$$I_0/I_t = e^{kCL} \tag{2}$$

where $C$ is the concentration of fluorophore, $L$ is the optical path length in the detecting volume and $k$ is the absorption coefficient. Substitute Equation (2) into Equation (1),

$$F = \beta I_0(1 - I_t/I_0) = \beta I_0 \left(1 - e^{-kCL}\right) \tag{3}$$

This equation can be simplified using Taylor's expansion of $e^{-kCL}$. When $kCL$ is very small,

$$e^{-kCL} \approx 1 - kCL \tag{4}$$

Substitute Equation (4) into Equation (3),

$$F = \beta I_0 kCL = QI_0CL \tag{5}$$

where $Q$ is the fluorescence constant related to the fluorophore, defined as the product of fluorescence quantum yield $\beta$ and the absorption coefficient $k$. Therefore, the fluorescence signal $F(t)$ measured by the PEC can be derived using

$$F(t) = \alpha F = QMI_0C \tag{6}$$

where $\alpha$ is the photoelectric conversion efficiency and $M = \alpha L$ is the module's system parameter, which is only relevant to the photoelectric structure parameter. Obviously, the measured fluorescence signal $F(t)$ is associated with only the intensity of excitation light $I_0$ and the concentration of fluorophore $C$ once the module's parameters are determined. Meanwhile, the measured fluorescence signal is linearly correlated with the concentration of fluorophore under stable intensity of the excitation light. Figure 1b shows the raw fluorescence signal converted by the analog digital converter.

## 2.2. Integrated LED-Induced Fluorescence Module

The integrated LED-induced fluorescence module is illustrated schematically in Figure 2a. The blue light emitted from the LED is projected into the sample consecu-

tively through a collimating lens, a homogenizer, a filter-2 and a dichroscope. In this fluorescence configuration, the collimating lens gathers and collimates the excitation light and the homogenizer modulates it into a uniform light column. The homogenizer is used to improve spot quality to eliminate measurement error caused by the non-uniformity of the excitation light $I_0$. A uniform light spot, 6 mm in diameter, is formed. The light column, whose spectrum is narrowed by the filter-2, is reflected at 90 degrees by the dichroscope. The uniform light spot irradiates fluorescent molecules in the sample and the ensuing light is emitted at longer wavelengths. Part of the emission light passing successively through the dichroscope and filter-1 is focused onto the APD's detecting window by the focusing lens. The dichroscope is used to separate the excitation path and emission path and to eliminate scattered excitation light in the propagation direction. Furthermore, the attenuator, which has a rough inner surface, is mounted at the end of the excitation path to eliminate scattered reflection. Our module configuration did not include the focusing objective lens used in the confocal fluorescence system, hence dramatically reducing the size.

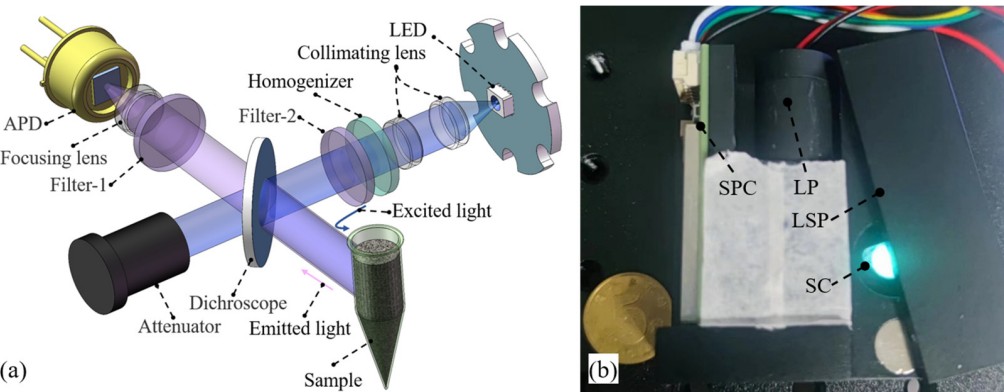

**Figure 2.** (**a**) Schematic view of the module configuration. APD: Avalanche Photodiode; LED: Light-Emitting Diode. (**b**) Photograph of the module during the testing process. LP: Light Path; SPC: Signal Processing Circuit; LSP: Light Shielding Plate; SC: Sample Cell.

In addition, the high integration of the LED excitation source is our fluorescence module's most important feature, as shown in Figure 2b. The integrated module is an elaborate fluorescence system requiring optics, mechanisms, electricity and computations, and is only 65 mm × 55 mm × 35 mm in size. An LED with 0.5 W radiated power and 475 nm peak wavelength was used as the excitation source. Its spectral response is represented by the red curve in Figure 3. Filter-2, with a 20-nm bandwidth, was used to tailor the radiation spectrum of the LED to get the excitation light whose spectral response is represented by the blue curve in Figure 3. Then, the excitation light spot with a diameter of 6 mm was built in the SC, whose axis is about 8 mm away from the module's light outlet. The peak wavelength and full width at half maximum (FWHM) of the excitation light spot are 475 nm and 16.8 nm, respectively. Furthermore, the APD type is BPW21R (Vishay Semiconductors), whose wavelength of peak sensitivity and range of spectral bandwidth are 565 nm and 420–675 nm, respectively. The SPC is an integrated control unit that carries a 24-bit analog–digital converter (LTC2484, Linear Technology Corporation, Milpitas, CA, USA) and MCU (STM32L010K4, ST Microelectronics, Geneva, Sweden). The MCU is responsible for signal processing and transmission, and a UART (maximum baud rate: 115,200 bps/s) is used for data exchange. LP is a tubular structure that is used to arrange all optical devices along an axis. An LSP with two magnetic sheets is used to shield against environmental interference.

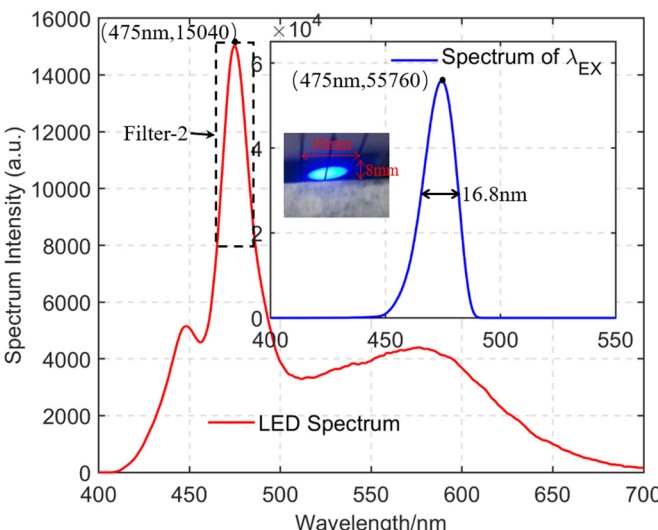

**Figure 3.** Spectral features of the LED and excitation light spot.

In conclusion, LED is used as the excitation source in the fluorescence module and has various performance advantages over lasers [13]:

(i)   Abundant selection of excitation wavelengths: the range of commercially available LED central wavelengths range from 255 to 4600 nm.
(ii)  Lower economic cost: LEDs often have a service life of up to 100,000 h at a cost of less than $1.
(iii) Small size, higher power, and straightforward operation: an LED is only a few millimeters in size, yet it can easily provide watts of output power with a constant drive current.

### 2.3. Target-Oriented Design Methods

The three-dimensional Excited-Emission Matrix (3D-EEM) fluorescence spectroscopy is one of the most efficient and commonly used techniques for analyzing fluorophore content [7]. However, the FS used in the 3D-EEM is bulky and time consuming, the lamp requires replacing regularly, and sampling is required for laboratory measurement and analysis. These are not suitable for automatic in situ analysis. Therefore, a compact device that enables rapid quantitative analysis of a specific fluorescent target is necessary. Here, a target-oriented design methodology guided by 3D-EFM is proposed. The 3D-EFM of lucigenin was obtained using an F280 steady-state fluorescence spectrometer (Tianjin Gangdong SCI.&TECH. Co., LTD, Tianjin, China), as shown in Figure 4a. The intersection between the red-dotted and pink-dotted lines in the 3D-EFM can be regarded as a point of separation between the excitation and emission paths. ① the selection rules for the best separation point should not only satisfy the peak wavelength of the excitation light, but also consider the dichroscope's transmission wavelength. In our module configuration, the dichroscope's transmission wavelength is 490 nm, which is a commercially available beam splitter. The dichroscope's transmission wavelength should be longer than the peak wavelength of the excitation light and their difference should be larger than 10 nm in order to eliminate scattered light from the sample cell. Lucigenin's emission curve under an excitation wavelength of 475 nm is shown in Figure 4b. ② the solid pink line defines the range of wavelengths entering the emission path. ③ The peak point in this range is regarded as the module's probe point. Thus the emission light, whose spectrum was narrowed by the line filter-1, is received by the APD. The choice of central wavelength and filter-1 bandwidth needs to be a synthetic tradeoff to maintain higher response sensitivity and avoid signal saturation, as shown by the red rectangle in Figure 4b.

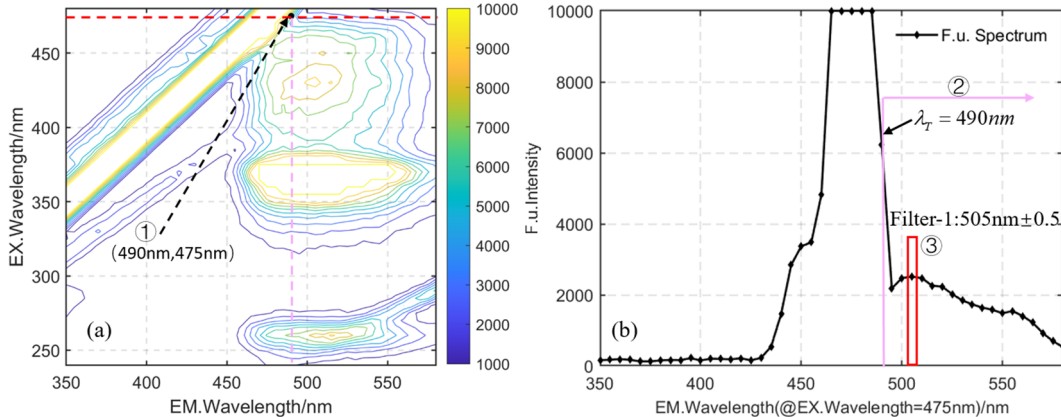

**Figure 4.** (**a**) 3D fluorescence scanning spectra of lucigenin (scanned range: 240–480 nm). (**b**) Lucigenin's emission curve under an excitation wavelength of 475 nm.

The optimized parameters for the integrated module are shown in Table 1. Based on the 3D fluorescence scanning spectra of lucigenin, filter-1's central wavelength does not match the optimal fluorescence emission wavelength that is used for distinguishing fluorescent molecules with overlapping emission spectra. Furthermore, the difference between the transmission wavelength of the dichroscope and the center wavelength of filter-1 should be greater than 10 nm. In addition, parameter optimization has to balance between sensitivity, detection limits and the optical device's manufacturing capacity. An integrated fluorescence detection module based on the lucigenin–chemiluminescence system was developed. This target-oriented design methodology possesses favorable extensibility and can enable rapid on-demand fluorescence-signal detection of other targets using an appropriate fluorescence reagent (Luminol or Sudan red et al.) and its parameter combinations.

**Table 1.** Integrated fluorescence module parameters for detecting lucigenin.

| Device | Parameter | Values |
|---|---|---|
| LED | Peak wavelength: $\lambda_0$ <br> Transmitted power: $P_{LED}$ | 475 nm <br> 0.5 W |
| Dichroscope | Transmission wavelength: $\lambda_T$ | 490 nm |
| Filter-1 | Central wavelength: $\lambda_1$ <br> Operation bandwidth: $BW_{f-1}$ | 505 nm <br> 1 nm |
| Filter-2 | Central wavelength: $\lambda_2$ <br> Operation bandwidth: $BW_{f-2}$ | 475 nm <br> 20 nm |
| AD | Digitalizing bit | 24-bit |
| APD | Peak sensitivity <br> Spectral range | 565 nm <br> 420–675 nm |
| MCU | Basic frequency | 32 MHz |
| | Flash memory | 16 Kbyte |
| | RAM | 2 Kbyte |

## 3. Experimental Section

### 3.1. Materials

Lucigenin (N, N-dimethyl-9,9′-biacridinium nitrate) [21,22] was used as a test sample in our experiments, as shown in Figure 5. Lucigenin is a well-known, commercially available chemiluminescence luminophore that has been wildly adopted to detect reactive oxygen species [23], Mn2+ ions [24] and Cl$^−$ ions [25]. Furthermore, it can emit bright fluorescence light at low concentrations, which means that it can be used as a good indicator of fluorescence calibration. Lucigenin's features of lower costs and excellent chemical

stability in aqueous solutions are especially suitable for in situ fluorescent hybridization testing. These advantages make it promising for sensitive analyses.

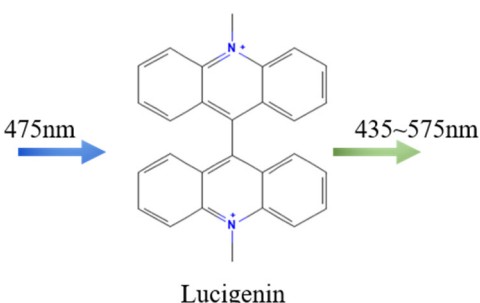

**Figure 5.** Schematic illustration of the photoluminescence strategy for detecting lucigenin activity.

### 3.2. Integrated Fluorescence Module Testing

To demonstrate the practicability of the integrated module, a lucigenin concentration detection experiment was carried out, as depicted in Figure 6. First, stock lucigenin solution with concentration of 1 mmol·L$^{-1}$ was diluted and divided into three portions labeled S1, S2 and S3. The lucigenin concentration in each sample varied from 0 μmol·L$^{-1}$ to 1 μmol·L$^{-1}$. The sample volume in each centrifuge tube was approximately 130 μL. The centrifuge tubes were successively placed into the sample cell. The sampling command was sent by the UART every 1 ms and the measured data were averaged over 10 ms. Finally, the average fluorescence intensity across 10 measurements was used as the observed value for the sample. The time required to measure a single sample using the integrated module was approximately 100 ms.

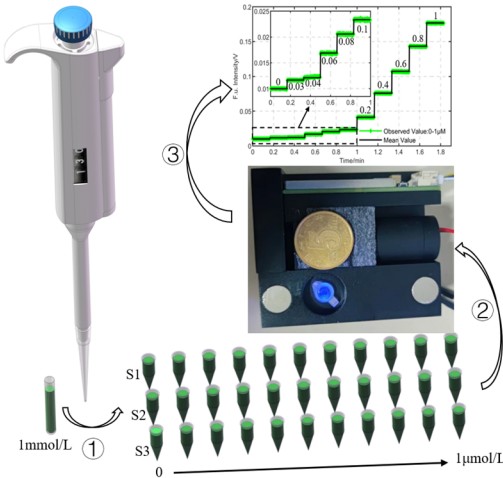

**Figure 6.** Schematic illustration of the experimental procedure.

### 3.3. Results and Discussion

Figure 7a–c illustrate the experimental results from three sets of samples. The time taken to measure each concentration point is 10 s, and the number of observed values recorded by the integrated module is 100. The average is shown by the solid black line in Figure 7a–c. It is obvious that the observed values of the three sample sets increased with the increase in the concentration of lucigenin solution. The integrated module's LOD are not higher than 0.03 μmol·L$^{-1}$. Figure 7d shows the linear relationship between observed values and lucigenin solution concentrations. The red dotted line is a theoretical measurement curve ($y = 0.1692x + 0.007$) calibrated in advance using concentrations ranging in value from 0 to 1 μmol·L$^{-1}$. We can see that all observed values are uniformly distributed along this curve and the sensitivity of the measurement is 0.1692 V/μmol·L$^{-1}$

in the concentration ranging from 0 to 1 µmol·L$^{-1}$. The non-zero intercept of the theoretical measurement curve indicates that the theoretical zero point of integrated module is 0.007 V. That is because module synthesis noise, including optical and electrical noise, cannot be completely eliminated. Therefore, the zero-point output of the integrated module is adjusted to 0.01 V to increase its resistance to disturbances. Theses experimental results indicate that our integrated module is highly consistent across multiple measurements.

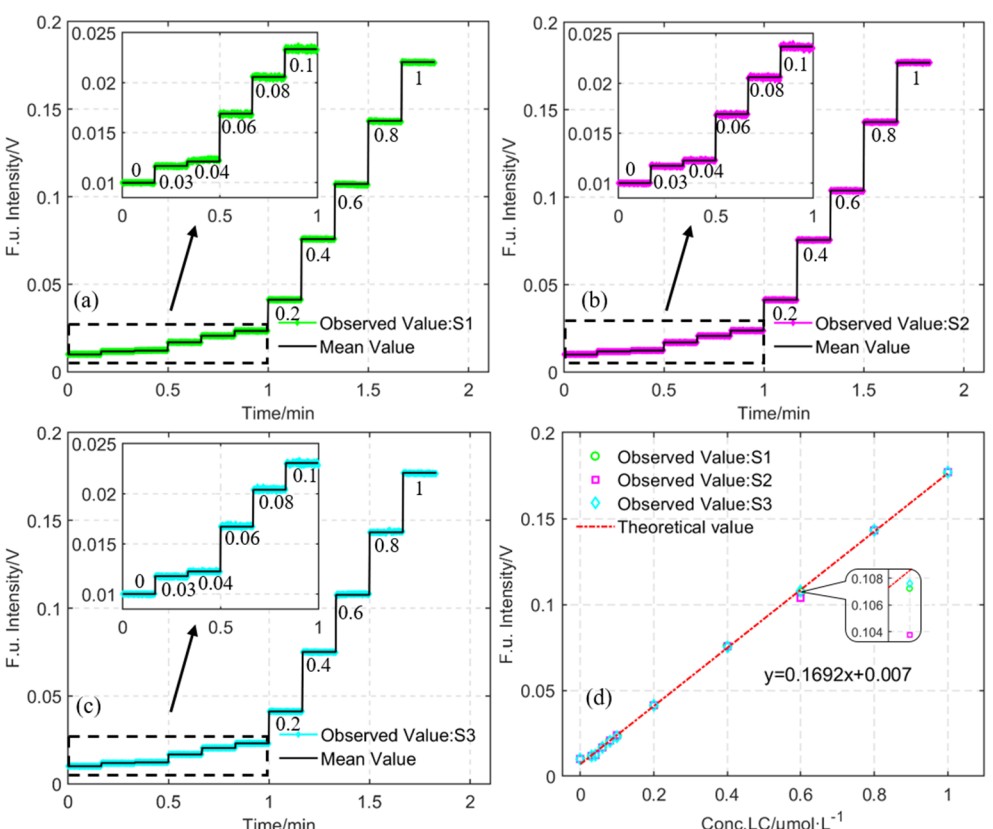

**Figure 7.** Results of lucigenin solution concentrations measured using the integrated module: Samples 1 (**a**), 2 (**b**), and 3 (**c**) are presented. (**d**) Linear association between lucigenin solution concentrations and values observed using the integrated module.

To demonstrate the integrated module's measuring precision, we recorded the values observed at 0.03 µmol·L$^{-1}$ and 0.2 µmol·L$^{-1}$, as shown in Figure 8. Each recording duration was 10 s. We see that the lower the lucigenin concentration in the detection volume, the flatter the observation. This is because the number of fluorescent molecules that go in and out of the detection volume is affected by random thermal motion. These data show that the observed errors in the values of the integrated module resulting from multiple measurements remain within 0.1 mV. Thus, when combined with the integrated module measurement curve, the effect coefficient of random thermal motion is no greater than 0.59 nmol·L$^{-1}$. Therefore, the integrated module has higher precision.

The experimental data were analyzed in order to evaluate the performance of the integrated module. The linear correlation between the observed values of the three samples and the sample concentrations was evaluated by computing Pearson's correlation coefficients. The calculation formula is as follows:

$$R_i = \frac{1}{N-1} \sum_{j=1}^{N} \left( \frac{x_{ij} - \hat{x_{ij}}}{\sigma_x} \right), i = 1, 2, 3; \ j \in [0, 1] \qquad (7)$$

where $R_i$ is Pearson's correlation coefficient of sample $i$, $N = 100$ is the number of repeated measurements, $x_{ij}$ and $\hat{x_{ij}}$ are the integrated module's observed value and theoretical value

in the *j* concentration of sample *i*, respectively. $\sigma_x$ is the standard deviation of $x_{ij}$. The measurement accuracy of the integrated module was evaluated using the RMSE whose formula is:

$$RMSE_i = \sqrt{\overline{(x_{ij} - \hat{x_{ij}})^2}} \tag{8}$$

where $RMSE_i$ is the root-mean-square error of sample *i*. The reliability of the integrated module measurement was evaluated using the MAPE, which is calculated using the formula:

$$MEAP_i = \left( \overline{\left| \frac{x_{ij} - \hat{x_{ij}}}{x_{ij}} \right|} \right) \times 100\% \tag{9}$$

where $MEAP_i$ is the mean absolute percentage error of sample *i*.

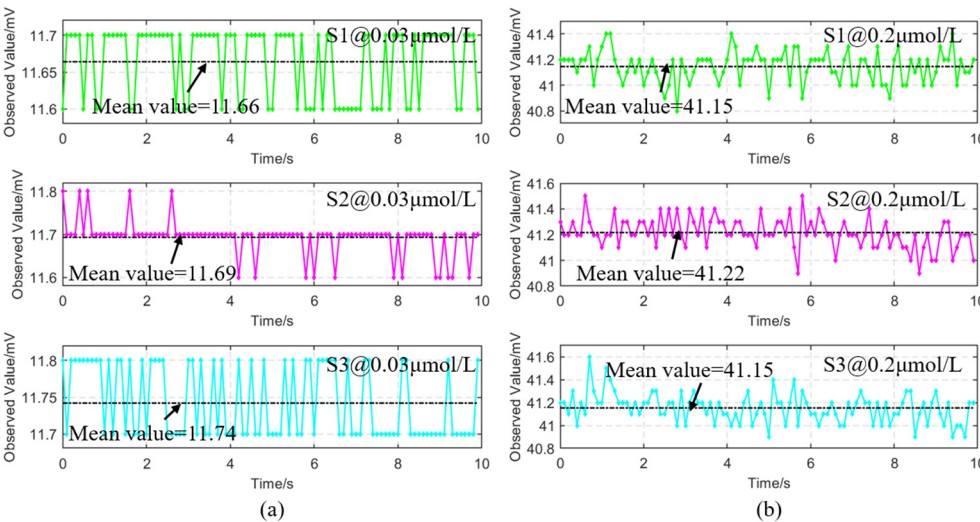

**Figure 8.** Errors in the measurements of three samples at (**a**) 0.03 μmol/L, (**b**) 0.2 μmol/L.

Table 2 shows the results of the statistical analyses. It indicates that the linear correlation coefficient between observed values and lucigenin concentrations is above 0.9995, and the RMSEs of multiple measurement results are lower than 1.8‰. Additionally, the MAPE of the sampling data is not more than 5.1048%, which is close to the reliable indexes of medical instruments. Therefore, this measurement model was shown to be effective and reliable.

**Table 2.** Performance metrics of the FCS module.

| Sample | R | RMSE | MAPE |
|--------|--------|--------|----------|
| S1 | 0.9998 | 1.2‰ | 5.1048% |
| S2 | 0.9995 | 1.8‰ | 5.0790% |
| S3 | 0.9998 | 1.2‰ | 5.0968% |

We conducted a comprehensive comparison between the integrated fluorescence module and other devices, as shown in Table 3. The LED-induced integrated module has several distinct advantages in terms of sensitivity, correlation and LOD compared with other fluorescence devices. Our integrated module has also been released from the constraints of a costly and bulky fluorescence detection equipment and is anticipated to leave the lab to meet more in situ detection needs. Despite the integrated module currently being limited to the detection of a single fluorescent chemical, module cascading or multi-color multiplexing [26] makes it possible to detect numerous fluorescent compounds simultaneously.

**Table 3.** Comparison between the integrated fluorescence module and other devices.

| Excitation | Sample | Performance indicators | | | Ref. |
|:---:|:---:|:---:|:---:|:---:|:---:|
| | | Sensitivity | Correlation | LOD | |
| LED | Lucigenin | $0.1692\ \mathrm{V/\mu M}$ | >0.9995 | $0.03\ \mathrm{\mu M}$ | * |
| LED | Chlorophyll-a | / | 0.9917 | $0.0025\ \mathrm{\mu g/L}$ | [9] |
| LED | DTT-Lucigenin | $3.59\ \mathrm{a.u./\mu g\cdot mL^{-1}}$ | 0.993 | $2.2\ \mathrm{ng/mL}$ | [21] |
| Xenon lamp | Tyrosinase | $1.04/\mathrm{\mu g\cdot mL^{-1}}$ | 0.9981 | $1.0\ \mathrm{\mu g/mL}$ | [22] |
| Laser | Glutathione | $2.61/\mathrm{\mu M}$ | 0.995 | 180 nM | [24] |
| Xenon lamp | Carbaryl | / | 0.93 | $9.2 \times 10^{-7}\mathrm{g/L}$ | [27] |
| LED | Amiloride | $23.3\ \mathrm{a.u./ng\cdot mL^{-1}}$ | 0.9997 | $1.43\ \mathrm{ng/mL}$ | [28] |

μM: $\mathrm{\mu mol\cdot L^{-1}}$; mM: $\mathrm{mmol\cdot L^{-1}}$; *: this study; DTT: 1,4-Dithiothreitol.

## 4. Conclusions

We have developed an integrated fluorescence module with an LED-excitation source and demonstrated a target-oriented module design methodology for measuring lucigenin concentration. The results proved that the FCS module has an outstanding linear response in the range of 0–1 $\mathrm{\mu mol\cdot L^{-1}}$, with sensitivity and LOD of 0.1692 $\mathrm{V/\mu mol\cdot L^{-1}}$ and 0.03 $\mathrm{\mu mol\cdot L^{-1}}$, respectively. Based on statistical data analysis, Pearson's correlation coefficient (R), root-mean-square error (RMSE) and mean absolute percentage error (MAPE) of the multiple-sample measurements were all above 0.9995, 1.8‰ and 5.079%, respectively. We believe that the FCS module has also been released from the constraints of a costly, bulky FCS equipment and is anticipated to leave the lab to meet more in situ detection needs.

**Author Contributions:** X.Q.: Instrument setup, conducting experiments, processing data, and writing the manuscript; M.Z., X.H. and L.J.: assisting with the experiments; W.G.: supervising the experiments; C.W.: project administration and supervision. All authors have read and agreed to the published version of the manuscript.

**Funding:** This work was supported by the Shandong Provincial Natural Science Foundation (Grants ZR2021QF033 and ZR2021QB019), the Postdoctoral Innovation Project of Shandong Province (Grant 202103002), the Qingdao Postdoctoral Applied Research Project (Grant 624200071311125), the National Natural Science Foundation of China (Grant 22104072), the Outstanding Youth Science Foundation of Shandong Province (Overseas) (Grant 2022HWYQ-058), the Southern Marine Science and Engineering Guangdong Laboratory (Guangzhou) (Grant GML2021GD0808) and the Shandong University Foundation for Future Scholar Plan.

**Institutional Review Board Statement:** Not applicable.

**Informed Consent Statement:** Not applicable.

**Data Availability Statement:** All data are available from the corresponding authors upon reasonable request.

**Conflicts of Interest:** The authors declare no conflict of interest.

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
