# Peer review of "Extensible LED-Induced Integrated Fluorescence Detection Module for Quantitative Analysis of Lucigenin Concentration"

_photonics, doi:10.3390/photonics10040392_

Round 1

Reviewer 1 Report

The manuscript presents a compact fluorescence spectrometer that utilizes an LED light source. While concentration measurements of lucigenin were conducted to validate the instrumentation, there are several areas that can be further improved to enhance the overall quality of the manuscript. Therefore, major revisions are recommended to address the following comments:

1.     The title of the manuscript refers to fluorescence correlation spectroscopy (FCS); however, the experiments only measure concentrations based on equation 4, which only utilizes G(0) and does not involve time-dependent correlations. Therefore, an additional experiment should be designed to incorporate autocorrelation or cross-correlation to demonstrate FCS. Without such an experiment, the manuscript would describe regular fluorescence spectroscopy.

2.     The manuscript lacks a clear justification for removing the objective before the sample in the prototype. Fluorescence emission is spontaneous and diverges rapidly once generated, but an objective placed near the emission source can collect and collimate the fluorescence to increase signal detection. Furthermore, a focused beam produces a very small confocal observation volume, which is necessary for statistical correlations to be observed. Thus, a clear justification should be provided for this design decision.

3.     The spectrum of the LED illumination, as well as the spectra of the excitation beam right before the sample, should be included in the supplemental information (SI).

4.     The experimental section should contain the brand and model names of all parts used to build the prototype. For example, the manufacturer and part number of the APD detector should be included.

5.     The manuscript should describe the data acquisition module, which is the instrument used to collect analog electrical signals from the APD. The sampling rate of the data acquisition should also be mentioned, as it determines the smallest time step in Fig 1b.

6.     The manuscript should describe how Fig 1b was obtained, including the sample and the spectrometer used for generating the plot.

7.     The manuscript should indicate the y-axis of Fig 3b.

Author Response

We appreciate very much your comments. Your suggestions will benefit the improvement of this manuscript and our future research. We would respond to all the comments and suggestions one by one.

Reviewer 2 Report

The authors presented article titled ‘Extensible LED-induced fluorescence correlation spectroscopy 2 integrated module for lucigenin concentration detection’ that has brought deep insights and developed an FCS module with high sensitivity and low limit of detection (LOD).  Over the year FCS turns to be a great tool for remote and even single-molecule detection. The response rate and LOD of the presented article are impressive. The use of LED over conventional laser is smart and cost-effective and the ability to use variable wavelength is a smart move.  Considering all I would suggest accepting the article in its present form. 

There are a few comments that can be added to improve the manuscript are mentioned below:

1. The spectral intensity and counts are very important for FCS to get some meaningful signal-to-noise ratio, the author should put more emphasis on the rationale behind LED over conventional laser source.   

2. Experimental details are missing such as instrumentation, model, and make in the current version.  Since it talks about the new instrumental design accurate representation is very important to readers. 

3. The detection limit and versatility can be further validated over to other systems. 

4. The accusation point in Fig 3b can be improved with a smaller step size.  

5. It would be encouraged to include the temporal time evolution spectral in the revised version. 

6. Since it is a custom build instrument, the authors should provide the data acquisition protocol/software details.

Author Response

(The authors gave the same response as above.)

Reviewer 3 Report

The manuscript is devoted to the development and testing of a portable luminescent analyzer. The authors obtained a prototype of the device and a statistically reliable linear calibration characteristic in studies of lucigenin concentration.

There are questions and comments on the text of the Manuscript:

1. Citations [3-7] and [14-17] should preferably be divided into 2-3 citations of 2-3 references.

2. In the Introduction, lines 68-76 are not appropriate, since they contain the results of this work. It is desirable to expand the literature review in the Introduction, including by analyzing analogues of this FCS from Table 3.

3. It is not advisable to start a section (for example, 2.1 or 3.2) right from the Figure. This is inconvenient for readers. Figures should be given immediately after the first mention of them, and not on the next page.

4. It is necessary to justify the use of a collimator, homogenizer, filters in the design of the device (Fig. 2a). Why is it impossible to excite the sample directly by LED radiation?

5. What are the spectral characteristics of the radiation source, radiation receiver, light filters? What are the parameters of the device components (other than those listed in Table 1)?

6. Line 144: Why compare the size of the device with the campus map?

7. Why is lucigenin chosen for testing the device? How can this fluorescent analyzer be used to monitor other substances or objects?

8. The choice of the excitation wavelength and luminescence spectrum according to Figure 3a is not clear. It is clearly seen that at λEXEM, the receiver will register the excitation radiation, which will have nothing to do with the sample. I can see from the picture that the excitation wavelength is about 360nm, and the range of light registration is about 460-550nm.

9. What physical quantity and in what units of measurement is represented along the ordinate axis in Figure 3b?

Despite the result obtained by the authors, I do not understand the rationale for choosing the excitation wavelength, the components of the device and the possibilities of using it to control anything other than lucigenin. After answering these questions, it will be possible to decide on the recommendation of the Manuscript for publication.

Author Response

(The authors gave the same response as above.)

Round 2

Reviewer 1 Report

Revisions are sufficient. No further comments.

Reviewer 3 Report

The authors have significantly corrected the Manuscript. Detailed answers to the questions are given.